# Hospital Caseload Demand in the Presence of Interventions during the COVID-19 Pandemic: A Modeling Study

**DOI:** 10.3390/jcm9103065

**Published:** 2020-09-23

**Authors:** Katsuma Hayashi, Taishi Kayano, Sumire Sorano, Hiroshi Nishiura

**Affiliations:** 1Kyoto University School of Public Health, Yoshida-Konoe-cho, Sakyo-ku, Kyoto 606-8501, Japan; katsuma5miffy@gmail.com (K.H.); kayano.taishi.2w@kyoto-u.ac.jp (T.K.); 2London School of Hygiene and Tropical Medicine, Keppel Street, London WC1E 7H, UK; sumire.sorano@gmail.com; 3CREST, Japan Science and Technology Agency, Honcho 4-1-8, Kawaguchi, Saitama 332-0012, Japan

**Keywords:** projection, hospitalization, policy making, epidemiology, mathematical model

## Abstract

A surge in hospital admissions was observed in Japan in late March 2020, and the incidence of coronavirus disease (COVID-19) temporarily reduced from March to May as a result of the closure of host and hostess clubs, shortening the opening hours of bars and restaurants, and requesting a voluntary reduction of contact outside the household. To prepare for the second wave, it is vital to anticipate caseload demand, and thus, the number of required hospital beds for admitted cases and plan interventions through scenario analysis. In the present study, we analyzed the first wave data by age group so that the age-specific number of hospital admissions could be projected for the second wave. Because the age-specific patterns of the epidemic were different between urban and other areas, we analyzed datasets from two distinct cities: Osaka, where the cases were dominated by young adults, and Hokkaido, where the older adults accounted for the majority of hospitalized cases. By estimating the exponential growth rates of cases by age group and assuming probable reductions in those rates under interventions, we obtained projected epidemic curves of cases in addition to hospital admissions. We demonstrated that the longer our interventions were delayed, the higher the peak of hospital admissions. Although the approach relies on a simplistic model, the proposed framework can guide local government to secure the essential number of hospital beds for COVID-19 cases and formulate action plans.

## 1. Introduction

The global pandemic of novel coronavirus (COVID-19) has unfolded since January 2020. Japan was one of the earliest countries to import cases. Globally, healthcare capacity has been the core issue to confront [1,2,3]: the expected incidence of severe cases admitted to intensive care units (ICUs) could well exceed the actual number of ICU beds available [4,5]. The epidemic intensified and was not kept under control in areas where hospitals were not able to manage an increasing number of cases (e.g., cities in Hubei Province in early January and north Italy in late February) [6,7]. As a result, when there was a rapid increase in the number of hospital-admitted cases in Japan in late March, tension was greatly elevated. Accordingly, the Japanese government and local prefectural governments decided to temporarily reduce the incidence of COVID-19 from March to May by closing host and hostess clubs, shortening the opening hours of bars and restaurants, and requesting a voluntary reduction of off-household contact [8]. Fortunately, hospital capacity and functions were maintained, and the incidence of hospital-admitted cases peaked in late April.

However, as the country successfully suppressed the incidence, it had to face another round of difficulty, that is, the second wave, with improved healthcare capacity for high caseload demand. Because the incidence greatly reduced by late May, there exists a gap of time before the next epidemic peak and a growing need to build up a system that helps to secure the essential number of hospital beds for the next waves of the epidemic [9]. Hospital admissions and severe cases act as a bottleneck in the epidemic response, and the predicted surge of healthcare demand could be used as a reference to determine the most appropriate timing (e.g., use the predicted peak date of the epidemic to determine the date on which the declaration of the state of emergency is made) of stringent interventions (e.g., request for self-restraining contact behavior) [10,11].

Since the first wave, it has become more feasible than before to offer more realistic, quantitatively sound hospitalization scenarios. In particular, it is possible to use first wave data in Japan to create projected scenarios. During the first wave, there were two notable characteristics: (i) there were more cases in the elderly in remote areas than urban areas, and (ii) the size of the epidemic was greater in urban areas than remote areas. The latter is a favorable finding for rural areas with limited hospital capacity, whereas the former is not good news for remote areas because the elderly population will require close medical attention. Moreover, older groups have been shown to be more infectious than younger groups [12]. Comprehensive and theory-based preparations would thus be of utmost importance.

The purpose of the present study is to project the number of cases and hospital admissions during the second wave using a simplistic mathematical model. Such modeling results may be useful for policy making in the healthcare supply system for COVID-19 cases for each local government.

## 2. Methods

### 2.1. Epidemiological Data

We analyzed the incidence data of confirmed cases by the date of illness onset from March to May 2020 obtained from open sources such as press releases and local government web pages. We substituted data for cases without a specified illness onset with the date of the report minus 9 days [13]. To capture age-dependent heterogeneity in an approximate manner, we stratified the epidemic curve into three age groups (i.e., 0–19, 20–59, and 60 years and older). Of these, the working-age population dominated cases in urban areas, whereas older people dominated cases in other areas. To capture the distinct pattern, Osaka Prefecture was analyzed to build the working-age centered model and Hokkaido to build the elderly centered model. In Japan, COVID-19 has been designated as an “infectious disease with special attention” according to the Infectious Disease Law, and cases are in principle mandated to be hospitalized.

### 2.2. Statistical Modeling

We divided the epidemic curve into two periods: before and after the declaration of the epidemic alert. In each period, we approximated the epidemic curve by exponential growth, and modeled the incidence of infection, i(t), as
(1)$$i\left(t\right)=\{\begin{array}{ll} i_{0}\exp\left(r_{1}\left(t-t_{0}\right)\right) & t<t_{1}\\ i_{0}\exp\left(r_{1}\left(t_{1}-t_{0}\right)\right)\exp\left(r_{2}\left(t-t_{1}\right)\right) & t\geq t_{1} \end{array},$$
where $i_{0}$ is the initial value, $t_{0}$ is the starting time of the epidemic, and $r_{1}$ and $r_{2}$ are the growth rates before and after issuing the alert, respectively. To fit the exponential growth well, we set $t_{0}$ to 15 January and 19 March 2020 for Osaka and Hokkaido, respectively. The first wave in Hokkaido existed from mid-February to 19 March, but we excluded the corresponding dataset for clarity. We set $t_{1}$ as the date on which the epidemic alert was issued and there was a reduction in incidence: the governor of Osaka Prefecture requested that the public avoid inter-prefectural movement and the government declared a state of emergency on 31 March. Hokkaido and Sapporo jointly declared a state of emergency on 12 April. As alternatives to the exponential curve, other parametric functions (e.g., logistic model) could be useful for capturing the cumulative incidence pattern [14]. However, the present study focuses on the absolute incidence to model the caseload demand in hospitals over time, to which exponential function suited the best, and moreover, the exponential curve allows us to convert the growth rates into reproduction numbers reasonably.

The incidence of infection for COVID-19 is not directly observable, while the illness onset event is readily observable. To fit the above i(t) to the illness onset data, we convoluted the probability density function of the incubation period with i(t). Then, the expected number of cases with illness onset, *c*(*t*), at time *t*, is
(2)$$c\left(t\right)=\int_{0}^{\infty }i\left(t-s\right)f\left(s\right)ds,$$
where f(s) is the probability density function of the incubation period with a mean of 5.6 days and a standard deviation of 3.9 days [15]. We assumed that the expected value *c*(*t*) followed a Poisson distribution and obtained the parameters that best fit the observed data using the maximum likelihood method.

To quantify the model, we also used the reporting delay from the illness onset to reporting to compute the daily number of newly reported cases. We also derived the reporting delay during the first wave in Japan from the literature, with a mean of 7.9 days and standard deviation of 5.5 days [6].

In addition to the maximum likelihood estimate, we computed a covariance matrix from the Hessian matrix, and performed parametric bootstrapping 10,000 times, assuming that each parameter was sampled from a multivariate normal distribution. Using a set of 10,000 parameters obtained from that process and also the Poisson distribution for each day, we obtained a stochastic realization of the epidemic.

### 2.3. Projecting the Possible Scenarios

Using the parameterized model, we simulated the possible second wave varying the timing of the declaration of the state of emergency. The threshold day, that is, the date on which the emergency is declared, was set to the day when the average number of confirmed cases per week exceeded 2.5 per 100,000 people [16]. The corresponding time $t_{1}$ was set to *a* day after the threshold, where the delay *a* is the assumed time gap for the decision-making process and ranges from 1 to 7 days. We assumed that the growth rate changes from $r_{1}$ to $r_{2}$ at time $t_{1}$ even during the second wave and as the number of secondary infections starts to decline.

As older people are more likely to suffer more severe symptoms [12], the epidemic curve was stratified by discrete age groups (i.e., children, young adults and older adults) so that we can assume that the hospitalization rate was different by age group [17]. We assumed that all older people must be managed in hospital facilities. It was also essential to incorporate the age-dependent pattern into our model because the age-dependent dynamics were markedly different between urban and remote areas (i.e., between Osaka and Hokkaido). Among other age groups with confirmatory diagnosis, we assumed that 30% of diagnosed patients in this age group would require inpatient treatment. This is because the proportion of patients admitted to hospital in Europe is on average 35% [18], and older people in Europe are also expected to be admitted most frequently. We calculated the number of inpatient beds assuming that all inpatients are discharged following a fixed admission length of 14 days [13]. Among admitted patients in hospital, we assumed that the risk of severe disease (i.e., ICU admission, ventilator use, or death) is 2.0% among children, 7.4% among working-age group and 13.4% among elderly, respectively [19]. We used these assumed values to calculate the number of critically ill patients.

### 2.4. Adjustment of the Reproduction Number and Patients by Age Group between Prefectures

We estimated the growth rate of the entire population and converted it into the reproduction number $R_{est}$ using the following formula [17]:(3)$$R={\left(1+rT\nu^{2}\right)}^{\frac{1}{\nu^{2}}},$$
where *T* is the mean generation time and *ν* is the coefficient of variation of the generation time [20]. Based on the probability density function of the serial interval, we assumed that the generation time distribution follows a gamma distribution with a mean of 4.8 days and a standard deviation of 2.3 days [21]. We used the relative reproduction number to adjust the growth rate for each age group using the following steps: (i) First, we calculated the ratio of $R_{i}$ = 1.4, 1.7, 2.0 to the baseline reproduction number
(4)$$R_{a,i}=\frac{R_{i}}{R_{{\mathrm{est}}_{\mathrm{total}}}}R_{a,est}\;i\in 1.4,1.7,2.0,\;j\in children,\;adults,\;eldery,\;R_{i}\in R_{1.4},R_{1.7},R_{2.0},$$
where $R_{a,i}$ is the “pseudo” reproduction number that corresponds to *R_i_* = 1.4, 1.7, 2.0 for the entire population using Equation (4). (ii) Second, we used the ratio to adjust the age-specific “pseudo” reproduction number obtained from Equation (3) using an age-specific exponential growth rate. (iii) Third, we substituted the adjusted reproduction number into Equation (3) again to calculate the required exponential growth rate for age group a, $r_{a,i}$. While age-specific growth rate is independently handled, the adjustment in Equation (4) makes it possible for age-dependent growth rates to be decreased by a common reduction rate in the reproduction numbers.

The reproduction number of 1.7 is the current status as of 29 May 2020. This value is consistent with the effective reproduction number observed in Tokyo and other cities in mid- and late-March. Moreover, the range of variation of 1.7 ± 0.3 is consistent with the scenario announced at the expert meeting on 2 March [13].

### 2.5. Adjustment to Each Prefecture

In addition to scenario modeling for Osaka and Hokkaido, we calculated the maximum number of hospital admissions for prefecture *j*, $M_{j}$, by adjusting the numerical results obtained above with respect to the population size. Using the calculated number of hospital admissions for the reference population (i.e., Osaka or Hokkaido) in age group *i*, $M_{i0}$, we adjusted the estimate by the ratio of the age-specific population sizes, $N_{0,i}$ and $N_{j,i}$, that is,
(5)$$M_{j}=\underset{i}{\sum }M_{0,i}\frac{N_{j,i}}{N_{0,i}}.$$

## 3. Results

In Table 1, we summarize the statistical estimates of the parameters used for modeling hospital admissions in urban and other areas. We adopt the working-age centered model in urban areas to reflect the dynamics of infection in metropolitan areas; otherwise, we adopt the elderly centered model. In both Osaka and Hokkaido, the growth rate before the declaration of the state of emergency was highest among children, followed by the elderly. Following the declaration, the growth rate among children and the working-age group greatly reduced.

Using the estimated parameters, we assessed the validity of our model using parametric bootstrapping. In Figure 1, epidemic curves obtained from each of the 10,000 simulations are added together with respect to time *t*, thereby confirming that the observed data points are mostly within the 95% confidence intervals of uncertainty for the epidemic curve.

Using the parameterized model, our subsequent process was to simulate a possible second wave, particularly the prevalence of hospital admissions. Considering the arbitrarily chosen threshold for issuing the alert during the first wave, we chose the threshold to be the cumulative number of cases during the past 7 days, which was 2.5 per 100,000 population or greater. Given that the population sizes of Osaka and Hokkaido were 8,848,998 and 5,304,413 persons, respectively, we assumed the threshold number of cases to be 221 and 133 per week. Figure 2 shows the epidemic curve of the prevalence of hospital admissions by age group, which could act as the basis for calculating hospital admissions. In Osaka, the peak of hospital admissions was dominated by young adults, whereas the highest peak in Hokkaido was for older people. The peaks were characterized by not only age-specific parameters leading to the growth rate (Table 1) but also the influence of different age-specific population sizes in those prefectures.

Additionally, we simulated varied scenarios in which the declaration of the alert was delayed for 1, 3 and 7 days from when the number of confirmed cases reached the threshold level for declaration. Figure 3 compares rather different trajectories by slightly different delays (see Appendix A). There are two notable characteristics. When the alert was delayed for 1, 3 and 7 days, the duration of the emergency state was 85, 90 and 101 days, respectively; that is, the duration of the emergency was extended for a longer period than the delay length of the declaration (Figure 3A–C). Moreover, and perhaps more importantly, the peak of hospital admissions was greatly elevated by the delay length of the declaration: for a delay of 1, 3 and 7 days, the peak was 2088, 2650 and 4272 cases, respectively. We used Equation (5) to anticipate the maximum number of hospital admissions during the second wave for all 47 prefectures in Japan (see Appendix A).

## 4. Discussion

The demand–supply imbalance of healthcare services for COVID-19 patients acts as a bottleneck in this epidemic [22]. In the present study, we projected the caseload demand by simulating an epidemic using a simplistic model while capturing the differential dynamics between urban and remote prefectures. Using an Osaka dataset, we parameterized a working-age centered model, whereas Hokkaido data were useful where older people dominated hospital admissions. We estimated not only the growth phase of the epidemic but also the reduced reproduction number according to the declaration of the state of emergency in these prefectures, which allowed us to simulate probable scenarios for the second wave.

The most important learning point gained from our exercise is that the duration of a hospital surge and peak of hospital admissions would be greatly eased by starting interventions at an early stage. Our finding is consistent with the published simulation study [23]. If the declaration of emergency is delayed for up to seven days, the duration is extended for a matter of a few weeks. Moreover, the peak of hospital admissions would be far greater than anticipated, and a delayed announcement would impose enormous pressure on healthcare capacities. These pressures would be experienced not only among healthcare facilities but also public health centers that were in charge of diagnosing cases and also tracing contacts. Thus, when and if the incidence reaches the threshold level to issue the alert, the declaration of emergency should be made at the earliest convenience.

As another learning point, our approach allowed different levels of urbanization to capture differential age-specific patterns of transmission. In urban areas, the focal group of heterogeneous transmission has been young adults, particularly those in their 20s–30s. During the second wave in Japan, the infection event in Tokyo was dominant in host and hostess clubs, and also drinking bars in a certain district [24]. By contrast, the tendency of the intense involvement of night time activities tapers in remote areas, and cases are more likely to be dominated by the elderly. Our model captures two such distinct patterns using age-specific growth rates of cases, and such scenarios would be applicable to similarly populated urban and remote areas.

An important strength of the present study is that it was successful in using actual COVID-19 epidemic data in the same country to parameterize the simple model. Compared with ordinary susceptible–infectious–recovered models, our proposed linearly (exponentially) simplified approach would be perhaps one of the most valid approaches to simulate hospital admission scenarios while taking advantage of a mixture of three exponential curves for each prefecture.

Limitations must be discussed. First, Osaka is the second-largest city in Japan, and Tokyo data were discarded. We made this choice because the linelist (i.e., case record) in Tokyo has not been shared openly. Despite this, we believe that it is reasonable to extrapolate the model of Osaka to other large cities because the age composition of the populations of Osaka and Tokyo are quite similar. Second, we chose Hokkaido to represent the aged society, but in fact, the capital city of Hokkaido, Sapporo, is dominated by young adults. In reality, it is vital to remember that clusters that are formed at nursing homes for the elderly can abruptly change the dynamics in remote areas. Third, we kept the model very simple, using even age-specific exponential growth rates to describe age-specific heterogeneities. If time allows, a more precise approach would be to quantify the age-dependent next generation matrix, and the matrix could highlight the differential impact of interventions by age. Fourth, our projection scenarios rely on the first wave in Japan. The Japanese government did not impose legally binding penal restrictions during the state of emergency, but a certain proportion of the population may not have behaved as the government wished. The next outbreak in Japan cannot be guaranteed to have the same effect without proper financial compensation from the government (i.e., the effectiveness of a similar declaration could be smaller).

Despite these issues, we believe that we have sufficiently quantified a model to allow each prefecture in Japan to anticipate the likely maximum demand of hospital admissions during a second wave using the first wave data. Additionally, to quantitatively project the likely peak of hospital admissions, it is vital that the next declaration is made in a timely manner, and such science-based decisions will be expected.

## Figures and Tables

**Figure 1 jcm-09-03065-f001:**
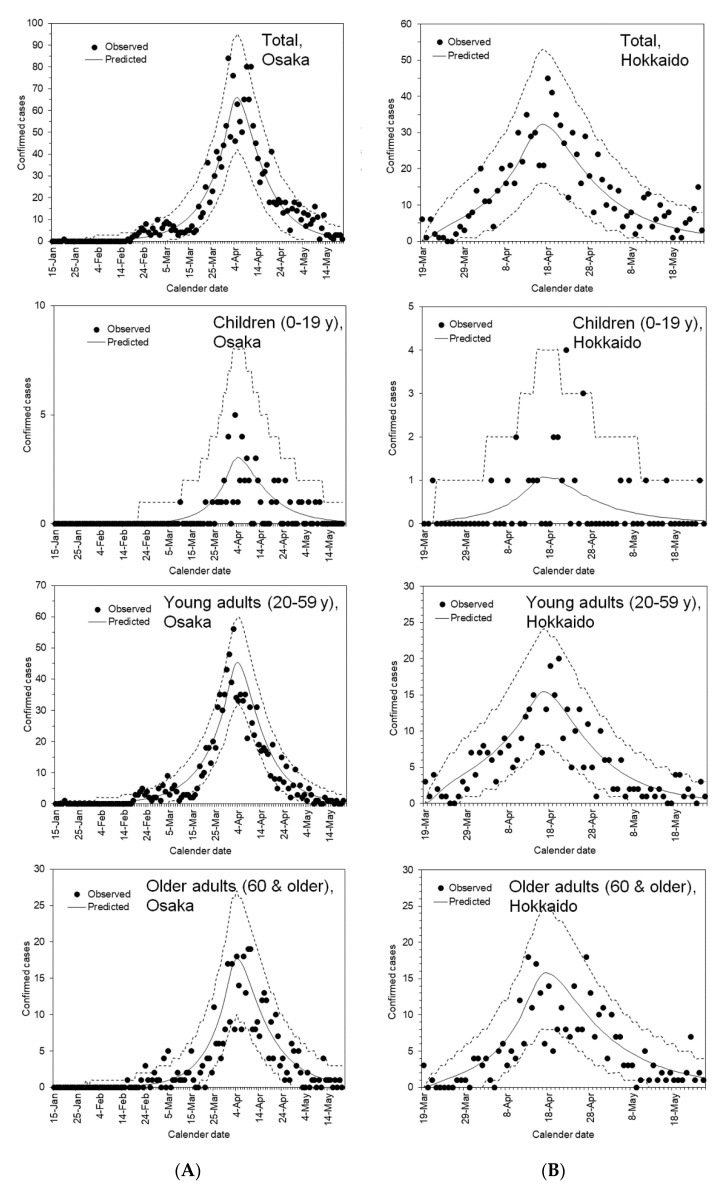
Comparison between the observed and predicted incidence of COVID-19 in Osaka and Hokkaido, 2020. Dots represent the observed incidence by the date of illness onset, whereas the solid line represents the fitted curves using maximum likelihood estimation and the dashed lines represent the 95% credible intervals derived from the parametric bootstrapping method; (**A**) Osaka Prefecture and (**B**) Hokkaido. Hokkaido was ahead of other prefectures in experiencing the first wave from early February. Following the comparison across all age groups on the top, comparisons among children (second from the top), young adults (third from the top) and older adults (bottom) are shown.

**Figure 2 jcm-09-03065-f002:**
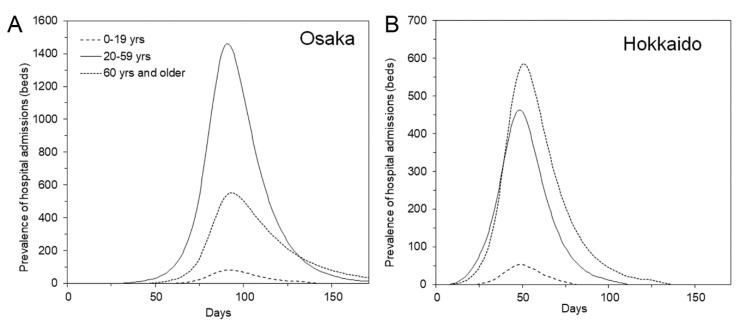
Reconstructed epidemic dynamics for working-age centered and elderly centered prefectures. (**A**) Working-age centered model, as parameterized by datasets in Osaka. The peak of hospital admissions of young adults was higher than that of older people. (**B**) Elderly centered model, as parameterized by Hokkaido data. The prevalence of hospital admissions of the elderly exceeded that of young adults. In both scenarios, we assumed that all confirmed cases were to be hospitalized for a fixed duration of 14 days.

**Figure 3 jcm-09-03065-f003:**
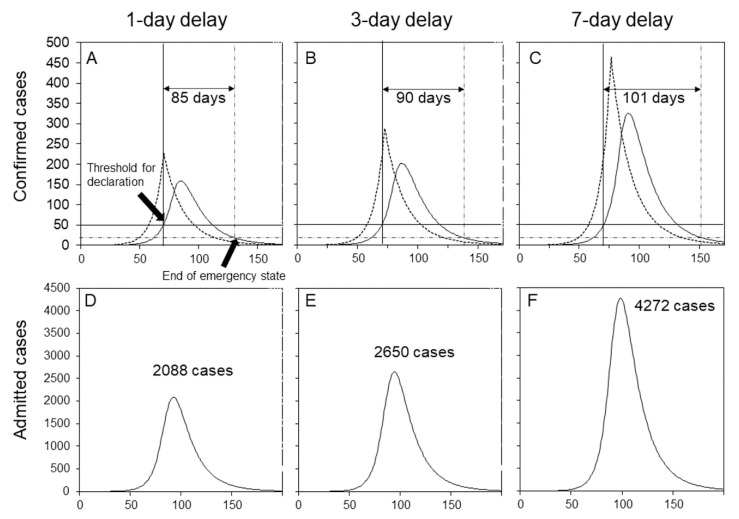
Intervention delay and scale of damage of the outbreak. (**A**–**C**) Dotted curves represent the number of cases by the date of infection (i.e., incidence of infection). Solid curves represent the number of cases by the date of reporting. The time point at which the solid line crosses the vertical and horizontal lines is when the threshold, that is, 2.5 per 100,000 people, is satisfied. The time point at which the vertical and horizontal lines cross is when the daily incidence is below 0.5 per 100,000. (**A**,**D**) One day after the threshold. (**B**,**E**) Three days after the threshold. (**C**,**F**) Seven days after the threshold.

**Table 1 jcm-09-03065-t001:** Estimated parameters for describing the dynamics of the model for hospital admissions.

	Osaka (Working-Age Centered Model)			Hokkaido (Elderly Centered Model)		
	i0	R1 *	R2 *	i0	R1 *	R2 *
All	0.0605	1.54	0.68	4.012	1.54	0.62
Children	0.0004	1.73	0.65	0.058	1.89	0.49
Young adults	0.0598	1.51	0.65	2.940	1.42	0.58
Older adults	0.0057	1.63	0.75	1.153	1.69	0.66

i0, initial value of the incidence; * R1, reproduction number before the declaration of the state of emergency; R2, reproduction number during the state of emergency. Note: R1 and R2 values by age groups are not strictly the reproduction number, but rather, were obtained using Equation (3) to convert the age-specific growth rate to a “pseudo” reproduction number.

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
