# Peer review of "Hospital Caseload Demand in the Presence of Interventions during the COVID-19 Pandemic: A Modeling Study"

_jcm, 2020, doi:10.3390/jcm9103065_

Round 1
Reviewer 1 Report
The authors first fit the first wave of SARS-CoV-2 using one example cities of Japan. The fitting routine is done using two regimes, the growing and the decaying. Each is fitted with a sum of three exponentials, where each exponential represents one of the age strata (young, adult, old). Second, they investigate how many hospital beds an outbreak of SARS-CoV-2 require, assuming an outbreak predominantly among the young or the elderly population. As expected, an outbreak among the elderly requires more hospital beds. Last, they show that the precise timing of mitigation measures strongly matters for the total number of beds required. Hence, early interventions might be of advantage.
I have a few comments.
- Please add the fit to the data from Hokkaido as well, to show that the fit of two exponentials is reasonable not only for the Osaka case numbers. Ideally, please choose a third city our region as well, e.g. Sapporo.
- In section 2.3, please revise the absolute numbers that have been chosen. These numbers are now known much more precisely. Ideally, add a few more references and report the uncertainties. Moreover, please clarify whether the proportion of cases is relative to reported cases, or relative to infections (case or infection related rats).
- In figure 1, please add the fits to each age group (potentially in the supplementary material).
- The result that the timing of an intervention (a few days earlier or later) has a large impact on the peak and the total number of cases, has been shown for SARS-CoV-2 by Dehning et al., Science 2020. Please refer to that in your discussion.
Author Response
Point-by-point responses: Managing hospital capacity with interventions during the COVID-19 pandemic: A modeling study (jcm-923918)
[Responses to the Reviewer 1]
The authors first fit the first wave of SARS-CoV-2 using one example cities of Japan. The fitting routine is done using two regimes, the growing and the decaying. Each is fitted with a sum of three exponentials, where each exponential represents one of the age strata (young, adult, old). Second, they investigate how many hospital beds an outbreak of SARS-CoV-2 require, assuming an outbreak predominantly among the young or the elderly population. As expected, an outbreak among the elderly requires more hospital beds. Last, they show that the precise timing of mitigation measures strongly matters for the total number of beds required. Hence, early interventions might be of advantage. I have a few comments.1. Please add the fit to the data from Hokkaido as well, to show that the fit of two exponentials is reasonable not only for the Osaka case numbers. Ideally, please choose a third city our region as well, e.g. Sapporo.
>>
We have revised the Figure 1, showing fit to both Osaka and Hokkaido. Moreover, the fit to all three age groups are shown in the revised Figure 1.
- In section 2.3, please revise the absolute numbers that have been chosen. These numbers are now known much more precisely. Ideally, add a few more references and report the uncertainties. Moreover, please clarify whether the proportion of cases is relative to reported cases, or relative to infections (case or infection related rats).
>>
We thank the reviewer for this comment. The risk of severe disease has been updated (P3L118-123). We have clarified that (i) 30% admission rate was conditioned on confirmatory diagnosis (P3L123), and (ii) the risk of severe manifestations was conditioned on hospital admitted cases (P3L128).
- In figure 1, please add the fits to each age group (potentially in the supplementary material).
>>
As mentioned above, Figure 1 was revised following the reviewer’s comments.
- The result that the timing of an intervention (a few days earlier or later) has a large impact on the peak and the total number of cases, has been shown for SARS-CoV-2 by Dehning et al., Science 2020. Please refer to that in your discussion.
>>
Suggested paper has been added to the reference (ref. 23) and discussed in P7L233.
Reviewer 2 Report
The authors have presented a detailed and carefully analysed model. They have nicely also shown the data as supplementary upon which some aspects of the model were based/data used. There is a great learning curve with Covid, so I am sure on one level we do need a wide range of analyses like this.
However, I felt there was lack of clarity in several respects. This confusion on my part may be because authors needed to make the paper succinct, or it may be because they are trying to do 'too much' with their analysis: so I was left feeling more focus would have helped.
1. the title, introduction and discussion mention hospital capacity. However, I do not clearly see this aspect modelled. There is a set of panels to Figure 3 presenting n of cases hospitalised, but I would contend this is not the same as hospital capacity. Of course, it indicates the demand on hospital services, but that is only one side of the question - even if we can estimate what our likely demand in n of cases is to be, at what level do we set capacity? The point is that capacity can be measured in various ways - eg, the time taken to deal with a case, not just the n of cases - so I cannot clearly see how this data helps in this regard? Perhaps it is a case of managing expectations, and the authors might more usefully entitle this paper around 'hospital caseload demand' or similar? In this regard, I offer an overview analysis (see: Anaesthesia. 2020 May 21:10.1111/anae.15144). This is not a detailed mathematical analysis, but simply an overview which at least outlines how to view 'capacity' in this context.
2. their model tries to several things it seems, and I was left unsure that it did all of these equally well. First (as discussed above) it seems to try help set hospital capacity. Second, it looks at age-related subsets. Third, it looks at geographical variation (urban vs other)...I feel all this unnecessarily complicates the model (I am unconvinced by the value of the approach outlines in Eq 4 for example), whilst not yielding any particular advantage (eg, even if we could predict age-related variation in demand, how would that help guide our capacity-setting, which really has to be quite generic and not targetted in this way)? I wonder if a simpler model, looking at overall numbers would be equally useful. Thus, the authors could present two versions: a simpler model and then the more complex one taking these subsets into account, and assess which is better. Another aim also appears to be identifying the time at which preventive interventional measures should best be taken. This is yet another aim of this model, making at least 4 aims in total (but see also second wave issues below, possibly making five aims).
3. As to the model details itself, I have the following areas of uncertainty/confusion:
(a) they modelled exponentially, but would alternative models such as logistic (or even logistic map) have been even more accurate? I mention here another overview (see: Anaesthesia. 2020 May 28:10.1111/anae.15151)?
(b) I do not understand where they say page 3 line 89 "The incidence of infection is not directly observable"...surely it is?
(c) the section on 'projecting the second wave' is not clear to me, ie, in terms of the precise equations being applied, because the authors do not refer back the equations being used, only to some of the parameter terms. Thus, a second wave is mathematically surely identical to the first, only it comes later - it is unclear from their explanation if their model explicitly predicts a second wave, or only if it is able to model one where one exists?
(d) I do not understand why they tried to model R. As an entity it is not necessarily relevant to their model (although arguably inextricably contained within it - see again Anaesthesia. 2020 May 28:10.1111/anae.15151). Expressed differently, it is unclear what this section adds to their model, over and above the equations previously presented?
4. Result line 198-204. Strictly, this section appears more Methods than Results. Yet, it is also an important part of the Methods - ie, to be able to predict things. Often in modelling there is a training set of data and then a test set, so this outlined here would seem to be the latter. Yet, when I view the supplementary material I do not clearly understand what is laid out - are these model predictions? and if so, then how do they relate to actually what happened? This is the aspect of the paper that might have been convincing to show how accurate the predictions are, but I cannot see this clearly.
In summary, while the effort in this analysis must be appreciated, this is a complex model that warrants clearer explanation at each stage, and may benefit from further simplification of aims. It is unclear how well predictions match real data. At one level these analyses remain important in a new disease, yet on another level, it is difficult to see how the conclusion is a novel one, since we do now know by various means that earlier lockdowns etc would have been beneficial.
Author Response
[Responses to the Reviewer 2]
The authors have presented a detailed and carefully analysed model. They have nicely also shown the data as supplementary upon which some aspects of the model were based/data used. There is a great learning curve with Covid, so I am sure on one level we do need a wide range of analyses like this. However, I felt there was lack of clarity in several respects. This confusion on my part may be because authors needed to make the paper succinct, or it may be because they are trying to do 'too much' with their analysis: so I was left feeling more focus would have helped. 1. the title, introduction and discussion mention hospital capacity. However, I do not clearly see this aspect modelled. There is a set of panels to Figure 3 presenting n of cases hospitalised, but I would contend this is not the same as hospital capacity. Of course, it indicates the demand on hospital services, but that is only one side of the question - even if we can estimate what our likely demand in n of cases is to be, at what level do we set capacity? The point is that capacity can be measured in various ways - eg, the time taken to deal with a case, not just the n of cases - so I cannot clearly see how this data helps in this regard? Perhaps it is a case of managing expectations, and the authors might more usefully entitle this paper around 'hospital caseload demand' or similar? In this regard, I offer an overview analysis (see: Anaesthesia. 2020 May 21:10.1111/anae.15144). This is not a detailed mathematical analysis, but simply an overview which at least outlines how to view 'capacity' in this context.
>>
We totally agree with the reviewer. We have not considered capacity management in a broad sense. The title of revised manuscript highlights “Anticipating hospital caseload demand in the presence of interventions” (P1). Reference to Pandit (2020) was added to Discussion (ref 22).
- their model tries to several things it seems, and I was left unsure that it did all of these equally well. First (as discussed above) it seems to try help set hospital capacity. Second, it looks at age-related subsets. Third, it looks at geographical variation (urban vs other)...I feel all this unnecessarily complicates the model (I am unconvinced by the value of the approach outlines in Eq 4 for example), whilst not yielding any particular advantage (eg, even if we could predict age-related variation in demand, how would that help guide our capacity-setting, which really has to be quite generic and not targetted in this way)? I wonder if a simpler model, looking at overall numbers would be equally useful. Thus, the authors could present two versions: a simpler model and then the more complex one taking these subsets into account, and assess which is better. Another aim also appears to be identifying the time at which preventive interventional measures should best be taken. This is yet another aim of this model, making at least 4 aims in total (but see also second wave issues below, possibly making five aims).
>>
We thank the reviewer for this comment. While there are seemingly four distinct aims, they are connected from each other: age-dependent dynamics was essential, because we handled two markedly different settings (i.e. urban and rural dominated by young and old, respectively). The equation (4) was used, because the equation allowed us to impose a common reduction rate of reproduction number to multiple (independently handled) growth rates by age group. Thus, while a simpler model without age would be far less complex, such model would not be practical enough to argue the age-dependent caseload demand.
The reviewer gave these comments, because these points were not smoothly understood. Thus, as a possible approach that we could sincerely take to improve the manuscript, we decided to improve writing on the associated points. For instance, the essential reason to stratify the epidemic curve by age group was given in P3L118-120. This is related to urban/rural issue and this point was also mentioned in P3L121-123. Moreover, the backup comment for equation (4) was provided in P4L145-147.
- As to the model details itself, I have the following areas of uncertainty/confusion:(a) they modelled exponentially, but would alternative models such as logistic (or even logistic map) have been even more accurate? I mention here another overview (see: Anaesthesia. 2020 May 28:10.1111/anae.15151)?
>>
We have highlighted that it is possible to model the cumulative incidence by other parametric functions (P2L89-P3L93), referencing to the suggested paper (ref. 14). The present study focused on the exponential curve due to its utility to model the absolute incidence (and thus, the caseload demand) and reasonable formula to be transformed the reproduction number.
(b) I do not understand where they say page 3 line 89 "The incidence of infection is not directly observable"...surely it is?
>>
We apologize for the confusion. We intended to state that the illness onset event is readily observable, while infection event itself is hardly observable. The corresponding sentence was rewritten accordingly (P3L94-95).
(c) the section on 'projecting the second wave' is not clear to me, ie, in terms of the precise equations being applied, because the authors do not refer back the equations being used, only to some of the parameter terms. Thus, a second wave is mathematically surely identical to the first, only it comes later - it is unclear from their explanation if their model explicitly predicts a second wave, or only if it is able to model one where one exists?
>>
We agree with the reviewer that the use of “second wave” could be confusing. We replaced it by possible scenarios (P3L110).
(d) I do not understand why they tried to model R. As an entity it is not necessarily relevant to their model (although arguably inextricably contained within it - see again Anaesthesia. 2020 May 28:10.1111/anae.15151). Expressed differently, it is unclear what this section adds to their model, over and above the equations previously presented?
>>
We thank the reviewer for this comment. This comment allowed us to understand that the earlier draft missed an important sentence why converting to the reproduction number is required. We apologize for the confusion. Accordingly, we have discussed in P3L145-147, clarifying that the common reduction rate can be applied to the reproduction number, not the exponential growth rate.
- Result line 198-204. Strictly, this section appears more Methods than Results. Yet, it is also an important part of the Methods - ie, to be able to predict things. Often in modelling there is a training set of data and then a test set, so this outlined here would seem to be the latter. Yet, when I view the supplementary material I do not clearly understand what is laid out - are these model predictions? and if so, then how do they relate to actually what happened? This is the aspect of the paper that might have been convincing to show how accurate the predictions are, but I cannot see this clearly.
>>
We entirely moved the corresponding part to Methods section (P4L152-158).
In summary, while the effort in this analysis must be appreciated, this is a complex model that warrants clearer explanation at each stage, and may benefit from further simplification of aims. It is unclear how well predictions match real data. At one level these analyses remain important in a new disease, yet on another level, it is difficult to see how the conclusion is a novel one, since we do now know by various means that earlier lockdowns etc would have been beneficial.
>>
We appreciate all comments from the reviewer. We carried out major revisions keeping in our mind that clear explanations are given to our model.
Round 2
Reviewer 2 Report
The authors have considered all the points of the reviewers carefully and I am persuaded they have met the requirements. I am also persuaded by the positive opinion and comments of Reviewer #1.
Perhaps unusually, my main concern relates the supplementary file in that I do not think an independent reader - nor I - can actually interpret this data. This section should be annotated properly to explain exactly what we are looking at, what the columns represent, etc. Therefore, the authors may need to convert this into a Word or pdf file rather than an Excel file to aid that understanding. I reailse that supplementaries are not part of the full paper, but they are highly relevant and useful to interested readers.
The other issue I have is with the title - it is too long. How about: "A modelling study of hospital Covid-19 caseload demand in Japan". Surely that says it all?
Author Response
Point-by-point responses: Managing hospital capacity with interventions during the COVID-19 pandemic: A modeling study (jcm-923918)
[Responses to the Reviewer 2]
The authors have considered all the points of the reviewers carefully and I am persuaded they have met the requirements. I am also persuaded by the positive opinion and comments of Reviewer #1. Perhaps unusually, my main concern relates the supplementary file in that I do not think an independent reader - nor I - can actually interpret this data. This section should be annotated properly to explain exactly what we are looking at, what the columns represent, etc. Therefore, the authors may need to convert this into a Word or pdf file rather than an Excel file to aid that understanding. I reailse that supplementaries are not part of the full paper, but they are highly relevant and useful to interested readers. The other issue I have is with the title - it is too long. How about: "A modelling study of hospital Covid-19 caseload demand in Japan". Surely that says it all?
>>
We thank the reviewer for this comment. We made the following revisions.
- Excel file was annotated as possible as we can. Because the Excel file can deliver the data directly to interested readers, we believe that we should maintain the file type. Instead, we made sure that the readers can interpret the table.
- We removed “anticipating” from the title and shortened the overall length.